# Right Heart Catheterization in Pediatric Pulmonary Arterial Hypertension: Insights and Outcome from a Large Tertiary Center

**DOI:** 10.3390/jcm11185374

**Published:** 2022-09-13

**Authors:** Zhuoyuan Xu, Hongsheng Zhang, Alexandra Arvanitaki, Chen Zhang, Qiangqiang Li, Bradley B. Keller, Hong Gu

**Affiliations:** 1Department of Pediatric Cardiology, Beijing Anzhen Hospital, Capital Medical University, Beijing 100029, China; 2Beijing Institute of Heart Lung and Blood Vessel Diseases, Beijing 100029, China; 31st Department of Cardiology, AHEPA University Hospital, 54621 Thessaloniki, Greece; 4Pulmonary Hypertension Service, Royal Brompton Hospital, London SW3 6NP, UK; 5Greater Louisville and Western Kentucky Practice, Cincinnati Children’s Heart Institute, Louisville, KY 40202, USA

**Keywords:** pediatric, right heart catheterization, pulmonary arterial hypertension, outcome

## Abstract

Aim: To define the clinical characteristics, hemodynamics, and adverse events for pediatric patients with pulmonary arterial hypertension (PAH) undergoing right heart catheterization (RHC). Methods: The large referral single center data of 591 diagnostic RHC procedures performed between 2005 and 2020 on pediatric PAH patients was retrospectively collected and analyzed. Results: A total of 591 RHC procedures performed on 469 patients with congenital heart disease (CHD)-PAH (median age 8.8 years, 7.9% New York Heart Association (NYHA) class > II, 1.5% with syncope) and 122 patients with idiopathic PAH (median age of 9.0 years, 27.0% NYHA class > II, 27.0% with syncope) were included. Of those, 373 (63.1%) procedures were performed under general anesthesia. Eighteen patients (18/591, 3.0%) suffered adverse events (mainly pulmonary hypertensive crisis, PHC, n = 17) during the RHC procedure, including 14 idiopathic pulmonary arterial hypertension (IPAH) patients and 4 CHD-PAH patients, and one IPAH patient died in hospital 63 hours after RHC. The risk of developing PHC was significantly increased in patients with IPAH (OR = 14.02, 95%CI: 4.49–43.85, *p* < 0.001), atrial blood gas pH < 7.35 (OR = 12.504, 95%CI: 3.545–44.102, *p* < 0.001) and RAP > 14 mmHg (OR = 10.636, 95%CI: 3.668–30.847, *p* < 0.001). Conclusions: RHC is generally a low-risk procedure in pediatric patients with PAH. However, PHC occur in approximately 3% of patients. Therefore, RHC should be performed in a large, experienced referral pediatric cardiology center, especially in pediatric patients with IPAH requiring general anesthesia.

## 1. Introduction

Pulmonary arterial hypertension (PAH) is defined according to the recommendation at the time of the study as a resting mean pulmonary artery pressure (mPAP) of 25 mmHg or above. According to the 5th World Symposium on Pulmonary Hypertension (WSPH), PH is divided into: group 1—pulmonary arterial hypertension (PAH); group 2—PH due to left heart disease; group 3—PH due to lung disease and/or hypoxia; group 4—PH due to chronic thromboembolic disease and other pulmonary artery obstructions; and group 5—PH with unclear and/or multifactorial mechanisms [1]. The distribution of etiologies for pediatric PAH is different to that of adults, with a greater predominance of idiopathic PAH (IPAH), PAH associated with congenital heart disease (CHD-PAH) and developmental lung diseases [2,3].

Right heart catheterization (RHC) is the gold standard for the measurement of pulmonary artery pressure and plays a vital role in the severity assessment of pulmonary vascular disease and the selection of therapeutic strategy [4,5,6]. RHC is generally considered a low-risk procedure, with a rate of complications of around 1% in adult patients [7,8]. However, pediatric patients tend to have higher risks due to low weight, respiratory depression, anesthetic sensitivity and relatively unstable hemodynamic status [9,10].

This study therefore set out to investigate the clinical characteristics and hemodynamics of Chinese pediatric patients with PAH from one of the largest pediatric PAH referral centers, and further, to quantify the safety and efficiency of RHC in this vulnerable population.

## 2. Methods

### 2.1. Study Patients

This was an observational cohort study of patients diagnosed with PAH from 3 months to 18 years of age who underwent baseline diagnostic RHC procedures at a large single tertiary referral center (Beijing Anzhen Hospital, Beijing, China) between January 2005 and December 2020. According to the “2015 ESC/ERS Guidelines on the Diagnosis and Treatment of Pulmonary Hypertension” [1], PH was defined as an increase of mean pulmonary artery pressure (mPAP) ≥ 25 mmHg at rest as assessed by RHC. PAH was defined by a pulmonary artery wedge pressure (PAWP) ≤ 15 mmHg and a pulmonary vascular resistance (PVR) > 3 Wood units in the absence of other causes of precapillary PH. IPAH was defined as PAH with no underlying disease. All research protocols were reviewed and approved by the local institutional ethics committee. Informed consent for research protocols and for RHCs was obtained from patients and/or parents/guardians as appropriate. If patients underwent repeated RHC, only the first procedure was considered.

The patients who could cooperate during the RHC procedure were examined under local anesthesia, while procedures in younger patients or patients with incompatible factors were performed under general anesthesia. The anesthetics used were ketamine, propofol, dexmedetomidine or opioids at the anesthetist’s discretion. Patients were not intubated for general anesthesia and oxygen masks were used as required, according to the patient′s condition. We performed arterial and venous blood gas tests during and at the end of RHC. The pH value included in this study was from aortic samples obtained during RHC.

Patients’ baseline demographics and clinical characteristics including primary diagnosis, New York Heart Association functional class (NYHA FC) [11], saturation of peripheral oxygen (SpO2), initial symptoms, time from onset of symptom to RHC, comorbidities, type of anesthesia, type of intervention, peri-procedural adverse events (hemoptysis, arrhythmia, thrombosis, arteriovenous fistula formation or pseudoaneurysm, heart and vascular injury, pulmonary hypertensive crisis) were collected and analyzed. Acute vasodilator testing (AVT) was performed with an initial dose of inhaled iloprost of 5 μg for 15 min (Ventavis; Bayer-Schering Pharma, Berlin, Germany), since nitric oxide is not available for medical use in China. Acute vasodilator testing (AVT) was performed in PAH patients whose mPAP > 40 mmHg without clinical features of left heart failure, and a positive response was defined according to current guidelines as a decrease in mPAP of at least 10 mmHg to <40 mmHg with a stable cardiac output [1,12,13]. Hemodynamics obtained from RHC procedures and responses to AVT were also collected. Patients diagnosed with complex CHD (e.g., univentricular physiology) or with left or right ventricular outflow obstruction were excluded; however, data collected following anatomic correction were included.

### 2.2. Statistical Analysis

Statistical analysis was performed using SPSS version 28.0 (IBM Corporation, Armonk, NY, USA). Normally distributed continuous data were summarized as mean ± SD; median (interquartile range (QR)) was reported when the distribution was not normal. Categorical variables were presented as numbers (percentage). Comparisons between groups were performed using the Wilcoxon rank sum test or the χ^2^ test, as appropriate. Logistic regression analysis was used to investigate the risk of pulmonary hypertensive crisis (PHC). A 2-sided *p*-value of 0.05 was considered to indicate statistical significance.

## 3. Results

### 3.1. Overall Characteristics

A total of 591 diagnostic RHC procedures were performed on pediatric PAH patients, including 469 on patients with CHD-PAH and 122 on patients with IPAH. These patients were from 27 provinces and regions of China. Overall, 357 (357/591, 60.4%) patients were female. A total of 373 (373/591, 63.1%) procedures were performed under general anesthesia, while the remaining patients who could cooperate during the procedure were investigated under local anesthesia.

### 3.2. CHD-PAH Cohort

The majority of RHC procedures were performed in patients with CHD-PAH (469/591, 79.4%). The median diagnostic age in patients with CHD-PAH at the time of RHC was 8.8 (3.3, 13.3) years; 37 (7.9%) patients were at NYHA FC III-IV at the time of RHC; 64.4% of patients were female; 296 (296/469, 63.1%) procedures were performed under general anesthesia, while the remaining cases were under local anesthesia. The median time from the onset of symptoms to the invasive establishment of diagnosis was 24.0 (6.0, 68.0) months. Clinical characteristics and hemodynamics of patients with CHD-PAH are presented in Table 1. Referral to RHC due to suspected PAH in the CHD group was due to abnormal physical examination incidentally without specific clinical manifestation in most of the patients (283/469, 60.3%). Other cases were detected by decreased exercise tolerance (62/469, 13.2%), recurrent respiratory infections (60/469, 12.8%), cyanosis (38/469, 8.1%), developmental delay (14/469, 3.0%), syncope (7/469, 1.5%) and hemoptysis (5/469, 1.1%). In addition, 6.4% (30/469) of patients were diagnosed with Downs syndrome, 4 had DiGeorge syndrome and 2 had Noonan syndrome.

Of the 414 patients with an uncorrected shunt lesion, 190 CHD-PAH patients underwent defect correction after RHC assessment, while 224 patients did not meet the indications for repair at that time. In addition, 55 patients were postoperative PAH who underwent RHC assessments. Figure 1 shows the distribution of defect correction status by age. The proportion of patients with postoperative PAH and non-correctable PAH increased with the increase of age. In our cohort, patients with a diagnostic age between 12 and 36 months were the group with the highest proportion of patients undergoing defect correction (74%).

Of all the CHD-PAH patients included, seven (7/469, 1.5%) had a history of syncope (Table 1) and five of them were female. One patient had suffered from at least two episodes of syncope. Their mean age at RHC was 14.5 ± 1.5 years, and RHC in all of them was performed under local anesthesia. Their mean mPAP was 68.6 ± 21.0 mmHg, their mean PVRI was 19.3 ± 12.4 WU·m^2^, their mean aortic pressure (AoMP)/mPAP was 1.7 ± 0.7 and their mean cardiac index (CI) was 3.4 ± 0.8 L/min/m^2^. Four patients were diagnosed with postoperative PAH. Two patients were with non-correctable ventricular septal defect (VSD) and atrial septal defect (ASD), respectively. One was an ASD patient. Notably, none of them experienced adverse events but responded poorly to iloprost during AVT.

As illustrated in Table 1, patients with postoperative PAH had the highest proportion of a history of syncope and NYHA-FC III-IV at the time of RHC. These patients with previous syncope also had higher right atrial pressures (RAP) and lower CI compared to CHD patients without syncope. Only four (0.9%) CHD-PAH patients suffered adverse events during RHC. PHC occurred in one patient with postoperative PAH. The other three adverse events occurred in non-correctable PAH patients, including two with PHC and one with bradyarrhythmia.

### 3.3. IPAH Cohort

A total of 122 RHC procedures were performed on patients with IPAH. The median age at RHC was 9.0 (5.0, 12.9) years. A total of 77 (63.1%) procedures were performed under general anesthesia, which was similar to the CHD-PAH patients (both 63.1%). In the IPAH group, 45.1% (55/122) of patients were female. The clinical characteristics and hemodynamics of patients with IPAH are shown in Table 2. Median time from onset of symptoms to catheter diagnoses was 12.0 (4.0, 25.5) months. A total of 33 (27.0%) patients were in NYHA-FC III-IV at the baseline diagnostic RHC time. Decreased exercise tolerance was the most common initial symptom for patients, which was found in 54 patients (54/122, 44.3%), followed by syncope, found in 33 patients (33/122, 27.0%). Of these, 17 had a history of recurrent syncope. Other cases were detected by shortness of breath (15/122, 12.3%), physical examination (10/122, 8.2%), edema (8/122, 6.6%) and hemoptysis (2/122, 1.6%). Compared with CHD-PAH patients, more IPAH patients had a prior history of syncope (27% vs. 1.5%, *p* < 0.001) and worse functional capacity (NYHA III-IV) (27% vs. 7.9%, *p* < 0.001). Compared to CHD patients, IPAH patients had a high risk of developing adverse events (n = 14, 11.5%). All adverse events were due to PHC during the procedures.

Syncope episodes were a common complication in IPAH patients in our cohort. The clinical characteristics and hemodynamics of patients with and without syncope episodes were compared, and syncopal patients tended to have higher mPAP [(73.1 ± 22.1) to (64.4 ± 24.5) mmHg, *p* = 0.007], older age at RHC [114.8 (61.8, 131.6) to 110.5 (60.2, 156.0), *p* = 0. 002] and higher PVRI [18.6 (11.3, 21.4) to 16.2 (9.3, 23.3) WU·m^2^, *p* < 0.001]. Notably, 11 of the 33 syncopal patients were AVT positive (33.3%), compared to 15.7% in the non-syncopal patients (*p* = 0.04).

### 3.4. Complications and Pulmonary Hypertensive Crisis

Eighteen patients (18/591, 3.0%) suffered adverse events during RHC, 14 in IPAH patients and 4 in CHD-PAH patients. Two patients were under local anesthesia and 16 were under general anesthesia. Seventeen (17/591, 2.9%) patients developed PHC, characterized by a sudden decline of systemic blood pressure with or without changes in heart rate. One CHD-PAH patient had arrythmias with III° atrioventricular block. Five patients underwent emergency cardiopulmonary resuscitation and three patients required unplanned endotracheal intubation. Among the 17 children with PHC, PHC occurred in five cases during induction of anesthesia, 10 cases during RHC, and two cases after pulmonary angiography. During the onset of PHC, there were seven cases of sinus tachycardia, three cases of sinus bradycardia and three cases of new-onset complete right bundle branch block. One case of short paroxysmal ventricular tachycardia, one case of high-grade atrioventricular block and two cases of ST-T segment dynamic changes were also reported. One IPAH patient died 62.5 h after the procedure due to refractory heart failure related to PHC. No patients died during the RHC.

Overall, in our study, PHC occurred more frequently in patients without an intracardiac shunt (14 in IPAH, 1in postoperative PAH). Clinical and hemodynamic characteristics in IPAH and CHD-PAH patients with PHC compared with non-PHC patients are shown in Table 3. The similarities in IPAH and CHD-PAH cohorts were that patients with PHC were younger, had lower arterial blood gas pH value during RHC procedures, had significant elevated pulmonary artery pressure (decreased AoMP/mPAP) and PVRI. These differences, however, were only statistically significant in the IPAH patients. More patients with PHC required noninvasive assisted ventilation or endotracheal intubation (both *p* < 0.001). Body surface area in IPAH patients with PHC was relatively smaller (*p* = 0.001) and RAP was higher (*p* = 0.004), and most PHC occurred under general anesthesia procedures (*p* = 0.014), while CHD-PAH patients have no significant differences in the above points (Table 3). In response to PHC, RHC was terminated in three patients as patients were felt to be too unstable to complete the RHC. Results of the logistic regression analysis, including odds ratio of PHC, for the entire cohort is shown in Table 4. Patients with IPAH, with lower atrial blood gas pH value and significant elevated RAP had significantly increased risk of developing PHC. Patients with decreased cardiac function, general anesthesia condition and had a history of syncope were also at higher risk with PHC.

## 4. Discussion

In our study, we reviewed clinical characteristics and hemodynamics of pediatric patients with PAH in a large, experienced single referral center in China. Eighteen (3.0%) patients suffered adverse events during RHC, 17 were PHC and one patient died in hospital. According to the PAH guidelines [1] and clinical practice, RHC is the core part of the hemodynamic diagnosis and evaluation of patients with PAH, but there are risks in performing RHC in children with PAH. According to the literature, the mortality rate of children undergoing RHC is 0.3% to 0.8%, but pediatric patients with PAH have a more than 10-fold increased risk of cardiac arrest [14]. CHD-PAH and IPAH were the most common types of pediatric PAH not only in China but all over the world [15,16]. Overall, our data reveal that RHC is a safe procedure in pediatric patients with CHD-PAH, while the probability of adverse events, especially PHC, was relatively higher in IPAH patients and correlated with poor cardiac functional class and poor hemodynamic condition, history of syncope, lower pH value and necessity of general anesthesia. However, with timely aggressive treatments, most pediatric patients can complete the procedures safely. These results suggest that RHC should be performed in an experienced large referral pediatric cardiology center, especially in pediatric patients with IPAH under general anesthesia [9].

Compared to patients with CHD-PAH, IPAH patients presented with similar levels of pulmonary artery pressure but higher PVRI. The time from onset of symptoms to diagnosis of PAH was shorter in IPAH patients than patients with CHD-PAH, indicating a more rapid disease progression, likely related to the different pathological mechanisms. In congenital correctable systemic-to-pulmonary shunts, the elevated pulmonary artery pressure is related to the volume/pressure overload of the pulmonary circulation and lesions tend to occur in arteries 100–200 μm in external diameter [6,17]. However, no certain etiology is clear in IPAH, and many patients even have a genetic background, with lesions tending to occur in arteries less than 100 μm [6,17] combined with poorer pulmonary vascular reactivity.

Previous studies have shown that pulmonary artery pressure higher than aortic pressure is an independent risk factor for PHC and other major complications in PAH patients under general anesthesia [9,18]. In our study, the mean AoMP/mPAP ratio was lower than 1 in patients with PHC both in IPAH and CHD-PAH cohorts. In addition, almost all patients with PHC have increased central venous pressure, even with significant increased RAP higher than 14 mmHg, suggesting severe right ventricular dysfunction. Right ventricular dysfunction and systemic hypotension are core factors to the occurrence and development of PHC [19]. All of the patients with PHC had different degrees of hypoxemia, and a small number of patients were confirmed to have respiratory acidosis with decreased pH value. Considering the lack of ventilation and the disturbance of the ventilation-perfusion ratio as the main reasons, non-invasive assisted ventilation or tracheal intubation was required.

In our patients, seven cases had sinus tachycardia, one case had short paroxysmal ventricular tachycardia, two cases had ST-T segment dynamic changes and three cases had new-onset complete right bundle branch block, which all suggested that the right ventricle overloaded, and suggested the requirements of timely and effective treatment. PHC can occur not only during RHC procedures, but also during the induction of anesthesia and after pulmonary angiography (n = 7, 41.2%), which indicates that pulmonary vasoconstriction caused by various inducing factors may also cause PHC, and these stimulating factors should be avoided as much as possible [19].

Syncope is a serious complication in patients with severe PAH. During presyncope, a sudden rise in PAP is accompanied by a gradual decrease in systemic arterial pressure, while during syncope, there is a simultaneous decrease in heart rate and in pulmonary and systemic arterial pressures [18]. Our study revealed that syncope mainly occurred in patients without shunts, including IPAH and postoperative PAH. Without the decompression effect of intra- or extra-cardiac shunts, circulatory collapse may be more prone to occur when PVR abruptly increases. This is the rationale for atrial septostomy and Potts-shunt for refractory PAH [4,20,21]. The proportion of patients in the current study with a history of syncope was significantly higher in IPAH, which may be related to the more serious pulmonary vascular lesions and PHC.

RHC is considered to be the gold standard for the measurement of pulmonary artery pressure and an effective tool to evaluate the pulmonary vascular reversibility [3,22]. According to the latest European Cardiology Society guidelines [1], defect closure was recommended in patients with PVRI less than 4 WU·m^2^. In our center, patients who underwent transcatheter closure followed the criteria mentioned above. However, the criteria for surgical operations were less restrictive, with significantly higher mPAP, PVR and PVRI, and lower SpO_2_. The reasons behind the tendency include: the size of defects was smaller in patients who were suitable for catheter intervention. Long-term follow-up studies are needed to evaluate the long-term prognosis of these patients.

RHC is generally considered a low-risk procedure, but severe complications can occur and can be fatal, including PHC, cardiac rupture, tricuspid valve injury, pulmonary artery rupture, and pseudoaneurysm formation [8,22]. Severe complications were reported to occur in about 1% of all RHC procedures [8,10]. In our cohort, 3% of procedures were aborted due to serious adverse events, and one patient died. Pulmonary hypertensive crisis, a potentially fatal complication, is seen when there is a rapid rise in PVR, leading to right heart failure and inadequate cardiac output [23]. The proportion of PHC was higher in IPAH patients and was relatively lower in patients with CHD-PAH, primarily occurring in patients with severe right ventricular dysfunction and systemic hypotension. Our study showed a clear trend towards the highest risk of complications being in patients with idiopathic PAH (13.5%) compared to 1.8% in CHD patients with postoperative PAH, 1.3% in CHD patients with PAH and uncorrectable defects, and 0% in those with PAH and correctable defects. Therefore, while none of the CHD PAH patients died, the risk of complications is highest in CHD patients with postoperative PAH.

While previous studies have reported on the morbidity and mortality risk of RHC in patients with pulmonary hypertension [8,14,24] not all of them were focused solely on pediatric patients, and to our knowledge, this is the largest study assessing morbidity and mortality as well as risk factors for adverse outcomes in a large contemporary Chinese population.

## 5. Limitations

We did not collect and further investigate the effects of different anesthetic drugs and depth of sedation on PHC and this should be attempted in further studies. Insufficient ventilation is also an important reason for PHC. However, patients in this study did not have dynamic monitoring of blood gas analysis indicators. It is, therefore, difficult for us to comment on the impact of ventilation status on PHC. Prospective studies need to be designed for further understanding.

Patients with correctable congenital heart disease have been included in the analysis, and these patients might not have irreversible pulmonary vascular disease. As the latter must be considered as a risk factor for complications related to RHC. Therefore, this suggests that complication rates might be higher than suggested by the current study in pediatric patients without significant pulmonary vascular disease.

This was a single center observational study including a large number of patients from a high-volume center in China. The results of this study cannot be directly extrapolated to low volume institutions or non-Asian populations.

In conclusion, our study reveals that RHC, as an invasive tool, is generally a low-risk procedure in pediatric patients with PAH. CHD-PAH and IPAH have different clinical characteristics, hemodynamic findings and risks during RHC. As PHC can be related to many factors, RHC should be performed in an experienced referral large pediatric cardiology center, especially in pediatric patients with IPAH under general anesthesia.

## Figures and Tables

**Figure 1 jcm-11-05374-f001:**
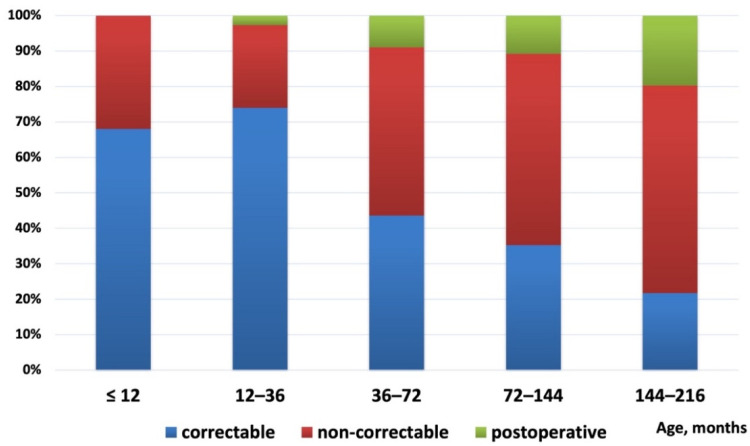
Distribution of defects correction status with patients age at the diagnostic right heart catheterization.

**Table 1 jcm-11-05374-t001:** Clinical characteristics and hemodynamics of patients with CHD-PAH.

	CHD-PAH	Correctable CHD-PAH	Non-Correctable CHD-PAH	Postoperative PAH
n	469	190	224	55
Diagnostic age, years	8.8 (3.3, 13.3)	4.4 (2.0, 10.9)	10.3 (5.3, 13.9)	12.4 (7.2, 15.0)
General anesthesia, n (%)	296 (63.1)	146 (76.8)	124 (55.4)	26 (47.3)
Female, n (%)	302 (64.4)	122 (64.2)	139 (62.1)	41 (74.5)
Height, cm	122.7 ± 32.5	107.7 ± 31.4	129.8 ± 30.1	142.6 ± 24.0
Weight, kg	27.1 ± 16.8	20.9 ± 14.9	30.1 ± 16.5	35.6 ± 17.2
Time from onset of symptom to diagnosis, months	24.0 (6.0, 68.0)	12.0 (4.0, 48.0)	24.0 (6.0, 84.0)	52.0 (14.3, 93.0)
NYHA-FC III-IV, n (%)	37 (7.9)	2 (1.1)	25 (11.2)	10 (18.2)
Symptoms before diagnosis				
History of syncope, n (%)	7 (1.5)	1 (0.5)	2 (0.9)	4 (7.3)
Development delay, n (%)	14 (3.0)	3 (1.6)	10 (4.5)	1 (1.8)
Hemoptysis, n (%)	5 (1.1)	2 (1.1)	3 (1.3)	0
RHC parameters				
mPAP, mmHg	65.2 ± 21.4	54.0 ± 21.0	74.5 ± 16.6	65.9 ± 22.1
RAP, mmHg	7.8 ± 3.0	7.4 ± 2.7	7.9 ± 3.0	8.9 ± 3.7
AoMP/mPAP	1.3 ± 0.5	1.6 ± 0.7	1.2 ± 0.3	1.4 ± 0.5
PVRI, Wood units·m^2^	12.4 (4.9, 20.7)	4.7 (2.6, 9.3)	16.8 (10.8, 24.4)	16.5 (8.9, 23.8)
SvO_2_, %	70.9 ± 8.3	72.1 ± 8.8	69.8 ± 8.2	71.5 ± 6.6
Qp/Qs	1.2 (1.0, 1.8)	1.8 (1.3, 2.5)	1.1 (0.8, 1.4)	1.0 (1.0, 1.0)
Rp/Rs	0.7 (0.4, 1.1)	0.4 (0.2, 0.6)	0.9 (0.7, 1.2)	0.9 (0.7, 1.2)
Cardiac Index, L/min/m^2^	4.3 ± 1.9	4.9 ± 2.2	3.9 ± 1.6	3.6 ± 1.1
Adverse events, n (%)	4 (0.9)	0	3 (1.3)	1 (1.8)

CHD-PAH = congenital heart disease associated with pulmonary arterial hypertension, NYHA FC = New York Heart Association functional class, RHC = right heart catheterization, mPAP= mean pulmonary artery pressure, RAP = right atrial pressure, AoMP = mean aortic pressure, PVRI = pulmonary vascular resistance index, SvO_2_ = mixed venous oxygen saturation, Qp/Qs = ratio of the pulmonary to systemic blood flow, Rp/Rs = ratio of pulmonary to systemic vascular resistance.

**Table 2 jcm-11-05374-t002:** Clinical characteristics and hemodynamics of patients with IPAH.

	IPAH
n	122
Diagnostic age, years	9.0 (5.0, 12.9)
General anesthesia, n (%)	77 (63.1)
Female, n (%)	55 (45.1)
Height, cm	129.7 ± 27.8
Weight, kg	31.8 ± 18.0
Time from onset of symptom to diagnosis, months	12.0 (4.0, 25.5)
NYHA-FC III-IV, n (%)	33 (27.0)
SpO2 at room air, %	97.3 ± 2.7
Symptoms before diagnosis	
History of syncope, n (%)	33 (27.0)
Short of breath, n (%)	15 (12.3)
Hemoptysis, n (%)	2 (1.6)
Edema, n (%)	8 (6.6)
RHC parameters	
mPAP, mmHg	65.7 ± 22.7
RAP, mmHg	8.5 ± 3.8
AoMP/mPAP	1.4 ± 0.6
PVRI, Wood units·m^2^	16.1 (10.0, 21.6)
SvO2, %	69.7 ± 7.4
Rp/Rs	0.8 (0.6, 1.2)
Cardiac Index, L/min/m^2^	3.5 ± 1.3
AVT positive, n (%)	25 (20.5)
AVT-PVRI, Wood units·m^2^	11.4 (5.9, 17.7)
Adverse events, n (%)	14 (11.5)

IPAH = idiopathic pulmonary arterial hypertension, NYHA FC = New York Heart Association functional class, SpO2 = saturation of peripheral oxygen, RHC = right heart catheterization, mPAP = mean pulmonary artery pressure, RAP = right atrial pressure, AoMP = mean aortic pressure, PVRI = pulmonary vascular resistance index, Rp/Rs = ratio of pulmonary to systemic vascular resistance, AVT = acute vasodilator testing.

**Table 3 jcm-11-05374-t003:** Clinical and hemodynamic characteristics in IPAH and CHD-PAH patients with PHC compared with non-PHC patients.

	IPAH (n = 122)			CHD-PAH (n = 469)		
	PHC	Non-PHC	*p* Value	PHC	Non-PHC	*p* Value
n	14 (11.5)	108 (88.5)		3 (0.6)	466 (99.4)	
Diagnostic age, years	5.5 ± 2.0	10.5 ± 0.7	0.002	6.7 ± 4.0	8.8 ± 5.4	0.610
Female, n (%)	8 (57.1)	47 (43.5)	0.335	2 (66.7)	300 (64.4)	0.934
Height, cm	115.6 ± 13.0	141.0 ± 6.0	<0.001	138.7 ± 37.1	122.7 ± 32.4	0.476
Weight, kg	18.5 ± 4.0	40.7 ± 6.9	0.006	34.9 ± 21.5	27.1 ± 16.8	0.612
Body surface area, m^2^	0.8 ± 0.1	1.1 ± 0.2	0.001	1.0 ± 0.5	1.0 ± 0.4	0.974
NYHA-FC III-IV, n (%)	5 (35.7)	28 (25.9)	0.438	1 (33.3)	36 (7.7)	0.101
History of syncope, n (%)	6 (42.9)	27 (25.0)	0.157	0	7 (1.5)	0.831
RHC parameters						
atrial blood gas pH value	7.29 ± 0.04	7.34 ± 0.02	<0.001	7.30 ± 0.06	7.37 ± 0.04	0.086
mPAP, mmHg	82.2 ± 11.8	65.0 ± 23.3	0.002	95.5 ± 23.2	65.1 ± 21.3	0.068
RAP, mmHg	11.3 ± 4.4	8.2 ± 3.3	0.004	7.6 ± 4.1	7.8 ± 3.0	0.806
PVRI, Wood units·m^2^	21.0 ± 5.3	16.6 ± 9.8	<0.001	26.7 ± 9.3	14.1 ± 10.6	0.128
AoMP/mPAP	0.84 ± 0.23	1.46 ± 0.59	<0.001	0.83 ± 0.16	1.34 ± 0.55	0.006
Anesthesia						
General anesthesia, n (%)	13 (92.9)	64 (59.3)	0.014	2 (66.7)	294 (63.1)	0.898
Duration of the procedure, hours	1.9 (1.2, 2.0)	1.5 (1.0, 1.6)	0.136	1.8 (1.0, 3.5)	1.5 (1.0, 2.0)	0.446
Natural airway, n (%)	11 (78.6)	106 (98.1)	<0.001	2 (66.7)	465 (99.8)	<0.001
Oxygen masks, n (%)	0	2 (1.9)	0.608	0	0	
Unplanned endotracheal intubation, n (%)	3 (21.4)	0	<0.001	1 (33.3)	1 (0.2)	<0.001

PHC = pulmonary hypertensive crisis, IPAH = idiopathic pulmonary arterial hypertension, CHD-PAH = congenital heart disease associated with pulmonary arterial hypertension, NYHA FC = New York Heart Association functional class, RHC = right heart catheterization, mPAP = mean pulmonary artery pressure, RAP = right atrial pressure, PVRI = pulmonary vascular resistance index, AoMP = mean aortic pressure.

**Table 4 jcm-11-05374-t004:** Odds ratio of pulmonary hypertensive crisis in the whole cohort.

Variables	Odd Ratio	95% Confidence Interval	*p* Value
RAP ≥ 14 mmHg	10.64	3.67–30.85	<0.001
atrial blood gas pH < 7.35	12.50	3.55–44.10	<0.001
NYHA FC III-IV	3.87	1.30–11.50	0.015
History of syncope	8.66	3.02–24.84	<0.001
Diagnosed with IPAH	14.02	4.49–43.85	<0.001
General anesthesia	4.56	1.03–20.13	0.045

RAP = right atrial pressure, NYHA FC = New York Heart Association functional class, IPAH = idiopathic pulmonary arterial hypertension.

## Data Availability

The data presented in this study are available on request from the corresponding author. The data are not publicly available due to data confidentiality reasons.

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
