# Peer review of "Right Heart Catheterization in Pediatric Pulmonary Arterial Hypertension: Insights and Outcome from a Large Tertiary Center"

_jcm, 2022, doi:10.3390/jcm11185374_

Round 1

Reviewer 1 Report

I have only one general comment which might strengthen the message of the paper:

The authors report data on patients with pulmonary hypertension related to congenital heart disease and data on patients diagnosed with idiopathic pulmonary hypertension.

The congenital heart disease group is subdivided into patients which have correctable and non-correctable heart disease and those which developed pulmonary hypertension after surgical treatment of their cardiac condition.

Patients with correctable congenital heart disease might be excluded from the analysis as many do not have irreversible pulmonary vascular disease. The latter, however, must be considered the risk factor for complications related to RHC. When excluding these patients without significant pulmonary vascular disease, complication rates will increase.

If data analysis and presentation are not altered, this aspect could be added to the discussion.

Author Response

We thank the reviewer for the supportive comments and suggestions. We fully agree with the reviewer that patients with CHD and correctable lesions have a lower risk of complications compared to those with uncorrectable lesions due to established pulmonary vascular disease. We have now mentioned this fact in the Discussion section and highlighted it in the Limitations of the paper as this is an important point for clinicians.

Reviewer 2 Report

In this paper Xu et al have the aim to define the clinical characteristics, hemodynamics, and adverse events for pediatric patients with pulmonary arterial hypertension  undergoing right heart catheterization. For this, ina single Center 591 diagnostic RHC,performed between 2005 and 2020 on pediatric PAH patients, was retrospectively collected and analyzed.

Xu et al. found adverse events only in 3% of patients, and conclude that RHC should be preferentially performed in a large, experienced pediatric cardiology center, especially in pediatric patients with IPAH requiring general anesthesia.

The number of patients included in the study is adequate, the data analysis correct and the paper is well written.

However, these are not completely original conclusions (not completely supported by results) and many similar articles are present in literature (bibliography could be completed), and do not increase our knowledge in the field.

No "limitations of the study" chapter is present and could be added.

Author Response

We sincerely thank the reviewer for the comments and suggestions. We have now followed the reviewer’s suggestion and have refocused the Discussion on the original findings supported by the data presented. We have also fully followed the reviewer’s suggestion and have added references to similar articles in the literature. Furthermore, an extensive Limitation section was added as suggested by the Reviewer.

We agree with the reviewer that previous studies have investigated the risk of adverse events in pediatric pulmonary hypertension catheterizations but strongly believe that our study adds to the literature by being the largest study from China whereas the majority of studies published so far were from Western countries. As the provision of care and the spectrum of patients differs substantially between these settings, we contend that the results of the current study are novel and should be of great interest the readership of the Journal. We have now added a paragraph to refer to previous studies and highlight the difference in our work.

Round 2

Reviewer 2 Report

The work has substantially improved in its present form and could be published